# Importance of tides and winds in influencing the nonstationary behaviour of coastal currents in offshore Singapore

Jun Yu Puah[1,2], Ivan D. Haigh[3], David Lallemant[1,2], Kyle Morgan[1,2], Dongju Peng[4], Masashi Watanabe[2], Adam D. Switzer[1,2]

[1]Asian School of the Environment, Nanyang Technological University, Singapore

[2]Earth Observatory of Singapore, Nanyang Technological University, Singapore

[3]School of Ocean and Earth Science, National Oceanography Centre Southampton, University of Southampton, Southampton, UK

[4]Department of Land Surveying and Geo-Informatics, The Hong Kong Polytechnic University, Hong Kong

*Correspondence to*: Jun Yu Puah (a210012@e.ntu.edu.sg)

**Abstract.** Coastal currents significantly impact port activities, coastal landform morphodynamics and ecosystem functioning. It is therefore necessary to understand the physical characteristics and natural variability of these currents within coastal settings. Traditional methods such as harmonic analysis assume stationarity of tide-driven currents and thus may not be applicable to systems modulated by variable nontidal inputs and processes. Here we deployed eight tilt current meters at shallow (< 5 m) coral reef environments in southern Singapore. Tilt current meters were positioned around the reefs at the main compass bearings to analyse the spatiotemporal variability of coastal currents in the frequency domain for one year (March 2018 to March 2019). Tidal motions were the primary mechanism of current flow on reefs and account for between 14–45 % of total variance across all sites, with diurnal currents having either similar or greater proportion of energy than semidiurnal currents. In Singapore, the diurnal wind stress, characteristic of the land–sea breeze, strengthens during the monsoons, and its effect on currents was investigated using wavelet coherence. Findings suggest that currents and wind stress were highly correlated at the diurnal and subtidal frequencies during the monsoons with a varying time lag of up to 6 hours with respect to both the phase and the antiphase. We find that wind forcing was responsible for the observed seasonal variations in the diurnal $K_1$ tidal constituent, its amplitude derived from short-term harmonic analysis. Given the importance of wind, we thus require longer time-series datasets to examine how atmospheric phenomena affect currents at greater time scales to improve predictions.

## 1 Introduction

Nearshore currents in coastal waters have a profound impact on various applications. Coastal communities rely on port operations, fisheries, and recreational tourism for economic growth and sustenance, and hence require robust infrastructure that are able to withstand strong currents (e.g. Klemas, 2011; Cervantes et al., 2015). Coastal currents also distribute nutrients,

pollutants and sediment either laterally or by vertical mixing, thereby not only influencing the morphodynamics of coastal landforms such as beaches, but also the composition and health of marine ecosystems such as coral reefs and mangroves (e.g. Covey and Barron, 1988; Klemas, 2011; Xue et al., 2012; Brunner and Lwiza, 2020; Sen Gupta et al., 2021). Therefore, it is imperative to improve our understanding of the physical characteristics of coastal currents and their natural variability across time and space.

Observed current velocities are the vector sum of the tidal and nontidal components. Nontidal currents are influenced by various factors, such as meteorological forcing and local bathymetry (e.g. Chen et al., 2014; Churchill et al., 2014). Tides and tidal currents can be expressed as a linear sum of sinusoidal tidal constituents, each with a known frequency (Foreman and Henry, 1989). Harmonic analysis (HA) is traditionally used to predict and fit the amplitude and phase of each tidal constituent to the observed tidal signal (e.g. Flinchem and Jay, 2000; Pawlowicz et al., 2002). For tidal currents, the lengths of semi-major and semi-minor axes (the latter representing the direction of rotation), phase and angle of inclination, collectively known as current ellipse parameters, are estimated. Cosoli et al. (2012) used HA on two years of surface current data in the northeastern Adriatic Sea and found tidal forcing to be weak, while correlation and coherence between wind forcing and nontidal currents are high. Churchill et al. (2014) also applied HA on Red Sea coastal currents over a two-year period and likewise found tidal currents to be weak. Nevertheless, both studies acknowledged the possible contamination of tidal velocities due to diurnal wind stress. This is because one key assumption of HA is that tidal currents are stationary over the entire record length, i.e. the amplitude and phase of each tidal constituent stay constant over time (e.g. Flinchem and Jay, 2000). This assumption is often violated due to external forcing such as variable wind stress, or other internal processes such as river flow and tide–surge interaction (e.g. Horsburgh and Wilson, 2007). Moreover, as tides propagate into shallower waters near the coast, they undergo distortion which may lead to the emergence of overtides, integer multiples of fundamental tidal harmonics which are typically used to describe nonlinear interactions such as friction and advection (e.g. Zhu et al., 2021).

Over the years, a variety of spectral techniques have emerged for studying nonstationary signals. The short-term harmonic analysis (STHA) involves conducting HA on short consecutive segments of the entire time series to provide moderate localisation of events involved in the modulation of tidal processes (Jay and Flinchem, 1995). Dusek et al. (2017) performed bimonthly HA on an 11-year record of estuarine currents in Tampa Bay, Florida, and found that tidal currents strengthen during periods of strong land–sea breeze and high freshwater discharge. However, STHA cannot resolve constituents with periods longer than the window length nor capture the temporal variation of tidal constituents (Flinchem and Jay, 2000). In contrast, the wavelet transform uses a finite function with translation and scaling parameters, also known as the mother wavelet, to build a class of functions with a finite integrated squared value (e.g. Flinchem and Jay, 2000). Specifically, the continuous wavelet transform (CWT) uses an arbitrary number of these basis functions to convert the time-series into two-dimensional plots of time and frequency, providing improved localisation of events (e.g. Jay and Flinchem, 1995; Torrence and Compo, 1998; Flinchem and Jay, 2000; Hoitink and Jay, 2016). However, the CWT is limited to extracting information on tidal species across

different frequency bands instead of examining specific tidal constituents such as $M_2$ and $K_1$ due to the time–frequency trade-off governed by the Heisenberg uncertainty principle (Torrence and Compo, 1998; Flinchem and Jay, 2000). Additionally, to measure the cross-correlation between two CWT, the magnitude-squared wavelet coherence (WC) is calculated (Grinsted et al., 2004).

The application of CWT is common in the study of estuarian dynamics. Jay and Flinchem (1995) revealed stronger semidiurnal and quarterdiurnal tidal signals during the generation of plume internal tides in the Columbia River Estuary, though their relationships with physical mechanisms remain ambiguous. Analysing the CWT of currents has also yielded insights on how seasonal river discharge influences tidal damping and the modulation of shallow water constituents in the Mahakam River in Indonesia (Sassi and Hoitink, 2013), the Yangtze River Estuary in China (Guo et al., 2015), and the Guadalquivir River Estuary in Spain (Losada et al., 2017). Zaytsev et al. (2010) employed both HA and rotary-multiple filter technique, the equivalent of CWT, and found persistent intense counterclockwise rotary currents presumably driven by counterclockwise sea breeze winds in the Bay of La Paz, Mexico. Currents have a greater propensity to nonstationary interactions than water levels and thus should not be overlooked (Guo et al., 2015). However, the relationship between currents and other drivers remains unclear, with local wind forcing having substantial impact on surface current variability (Ursella et al., 2006).

With this in mind, we aim to explore the nonstationary behaviour of coastal currents and their various drivers, with a focus on the Singapore Strait. The Singapore Strait links the South China Sea (SCS) and the Malacca Strait and is one of the busiest shipping routes in the world (e.g. Chen et al., 2005; Hasan et al., 2012). It has also undergone extensive development such as land reclamation and shore protection work that has majorly altered shoreline configuration and inevitably impacted local hydrodynamics. In the Singapore Strait, tidal asymmetry exists since tides transition from predominantly diurnal in the east to semidiurnal in the west (Chen et al., 2005; Tkalich et al., 2013), hence tides are semidiurnal while currents are mostly diurnal (Van Maren and Gerritsen, 2012; Peng et al., 2023). The tidal asymmetry is further complicated by the complex bathymetry as many islands such as rock outcrops and coral reefs lie within the strait (Chen et al., 2005; Hasan et al., 2012). Additionally, the hydrodynamics are affected by local wind systems. Yearly, Singapore experiences two monsoon seasons with two inter-monsoons in between: the northeast (NE) monsoon from mid-November to March when northeasterly winds blow across the SCS, and the southwest (SW) monsoon from mid-May to mid-September when wind circulation reverses (Van Maren and Gerritsen, 2012; Martin et al., 2022). The stronger and longer NE monsoon results in currents flowing net westward along the hydrodynamic pressure gradient from the SCS out to the Malacca Strait and Java Sea and only reverses in direction during the SW monsoon (Chen et al., 2005; Van Maren and Gerritsen, 2012; Martin et al., 2022).

Due to a lack of long-term observational records from Singapore, local hydrodynamics are typically modelled at coarse resolution and predicted using numerical models. Three-dimensional models have been used to simulate tidal currents in the Singapore Strait (e.g. Chen et al., 2005; Zhang, 2006), and further refined by expanding the grid boundaries (Hasan et al.,

2012; Van Maren and Gerritsen, 2012) and incorporating multi-scale nesting to improve grid resolution (Hasan et al., 2016). These models were forced by major tidal constituents and did not include wind forcing due to heavy computational demands, thereby making the study of hydrodynamics in the Singapore Strait especially challenging. Nevertheless, such studies are crucial given their potentially significant implications on port operations, coastal sediment dynamics, and larval connectivity of marine ecosystems. In particular, sedimentation of fine-grained sediment is chronic within Singapore Strait, and it influences the distribution and growth of coral reef communities. Here we focus on two coral reef platforms located within southern Singapore: Pulau Hantu (1.226247° N, 103.747049° E) and Kusu Island (1.225354° N, 103.860104° E) (Fig. 1). These fringing coral reefs are consistently subjected to high turbidity and sedimentation, as well as significant nearshore current velocities (Morgan et al., 2020). We used a one-year time-series of current velocity and direction data to achieve the following objectives: (1) analyse the spatial and temporal trends of raw current speeds and wind stress, (2) estimate the power spectral density (PSD) of currents in the Singapore Strait by partitioning the variance into four frequency bands and quantify the contribution of tides, (3) examine how the spectral properties of both currents and wind vary with time using CWT, and (4) investigate their relationship using WC, complemented with the results of STHA. The structure of the paper is as follows: Sect. 2 covers the type of data collected and the methods employed to achieve the study objectives. The results of the study are described and discussed in Sect. 3 and Sect. 4 respectively. Lastly, conclusions are given in Sect. 5.

## 2 Data and Methods

### 2.1 Field data collection

At Pulau Hantu and Kusu Island, we each deployed four Tilt Current Meters (TCM: TCM-1 Lowell Instruments) following the main compass axes of the reefs (north, south, east and west) (Fig. 1). TCMs operate by the drag-tilt principle, in which the meter tilts in the direction of the drag of the fluid. To minimise the effects of vortex eddies, turbulence and waves, the TCM was configured to record data at 8 Hz for 15 seconds (i.e. 160 samples) per minute and the data is burst-averaged over each minute (Lowell et al., 2015). The accuracy specifications of TCMs for speed and direction respectively are 3 cm $s^{-1}$ + 3 % of reading and 5° for current speeds exceeding 5 cm $s^{-1}$. We collected data on speed, direction, zonal (u-) and meridional (v-) velocities at a 10-minute sampling interval at a shallow depth of ~3 m from March 2018 to March 2019, except Hantu North where data were collected until December 2018 due to instrument malfunction. From the 51,000 data observations, there were eight and seven consecutive missing data points in Hantu West and Kusu South, respectively. Since data needs to be continuous for wavelet analysis, we interpolated the missing data using a moving median with a window length of 2 hours.

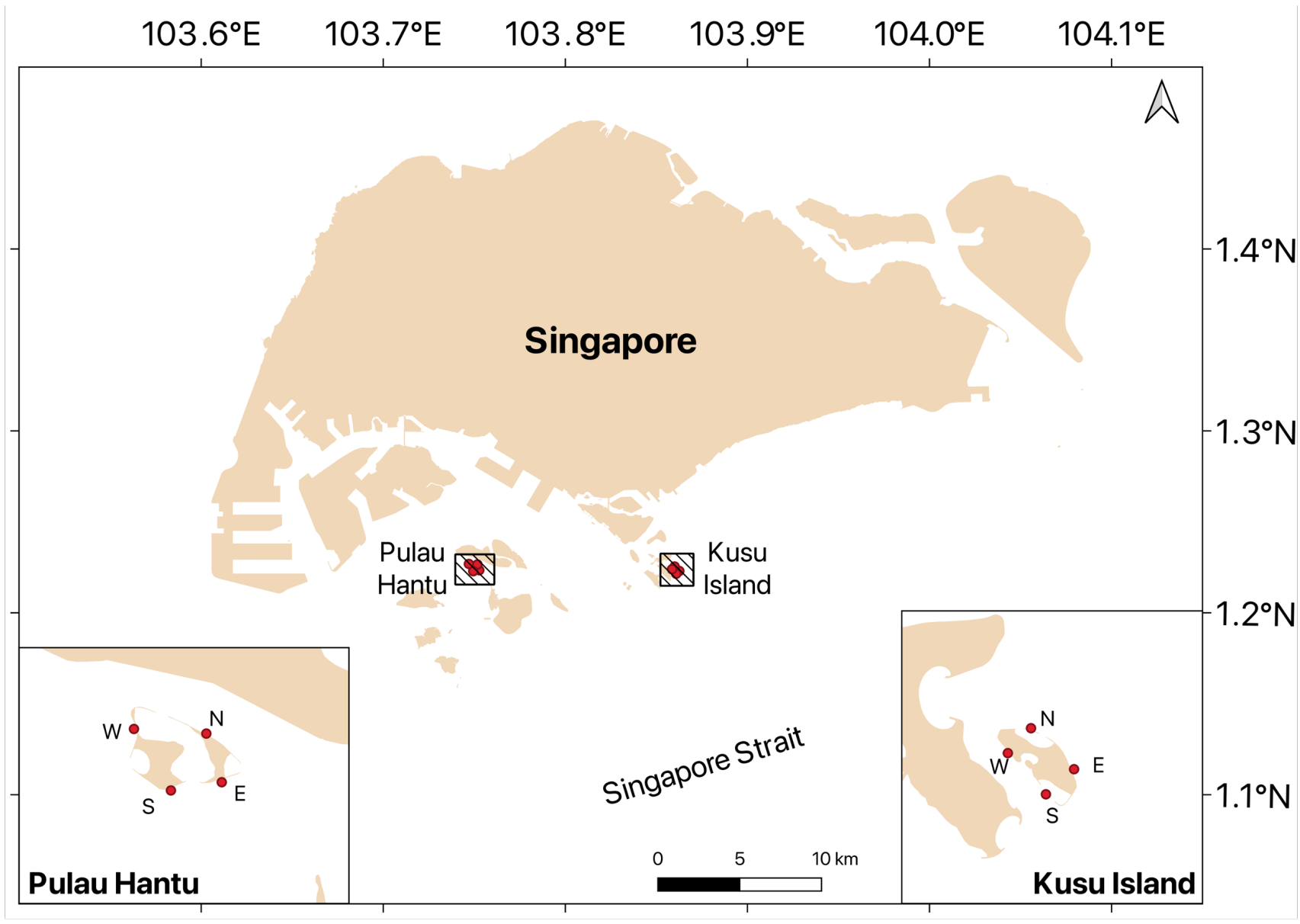

**Figure 1.** Location of Pulau Hantu and Kusu Island positioned off the southern coast of Singapore, indicated by the shaded boxes, and the eight study sites, indicated by the red dots on the inset maps.

To examine the role of local winds, we downloaded reanalysis data on zonal and meridional wind speeds at hourly resolution over the observational period from ERA5, the fifth generation ECMWF atmospheric reanalysis (Hersbach et al., 2020). Dynamical downscaling was performed by using the Weather Research and Forecasting (WRF) regional model forced by ERA5 reanalysis data and increased the spatial resolution from about 31 km, or 0.25 ° x 0.25 °, to 10 km horizontal resolution. We extracted the downscaled wind data from the grid points closest to both islands and used the drag coefficients from (Large and Pond, 1981) to derive wind stress from wind speeds at 10m reference height.

### 2.2 Analysis methods

We first conducted a preliminary examination of the raw speed of the currents and wind stress in the time domain. Currents flowing along the coastline are generally rectilinear (Thomson and Emery, 2014), and monsoonal winds blowing across the SCS are highly directional. Therefore, we performed Principal Component Analysis (PCA) to resolve both variables, initially expressed as eastward and northward vectors, into uncorrelated orthogonal major and minor axis components with their mean value subtracted. The greatest proportion possible of the total variance is thus explained by the major axis component. For currents, they represent the alongshore and cross-shore components respectively.

Using the MATLAB toolbox "jLab" (Lilly, 2019), we chose a time-bandwidth product of 3 and 13 Slepian tapers to obtain multi-taper PSD estimates of currents and wind. This method provides a smoother estimate of the spectrum than the traditional

Fourier transform as the spectrum is an average of multiple independent spectra estimates that are generated from the same sample, thereby minimising spectral leakage (Percival and Walden, 1993). We visually inspected the spectra for any peaks and gaps before partitioning the variance into 4 frequency bands: low frequency, $f$ < 0.72 cpd; diurnal, $f$ = 0.72–1.5 cpd; semidiurnal, $f$ = 1.5–2.3 cpd; and high frequency, $f$ > 2.3 cpd. Diurnal and semidiurnal frequencies hereafter will be collectively referred to as tidal frequencies. We then quantified the variance by integrating the spectrum over each frequency band and assessed their relative contribution. The 95 % confidence intervals for spectral estimates were calculated assuming a $\chi^2$ distribution of variance.

Wavelet analysis was done with the MATLAB toolbox "Cross wavelet and wavelet coherence" (Grinsted, 2023). To analyse the time series at different periods or frequencies, the wavelet is scaled accordingly and translated in time. We selected the Morlet wavelet as it is optimal for processing tidal signals, 12 scales per octave, and a dimensionless frequency of 6 to provide a good balance between time and frequency resolution (Torrence and Compo, 1998; Grinsted et al., 2004). The wavelet power is normalised by the variance of the time series, and the cone of influence (COI) demarcates the area where edge effects cannot be ignored (Grinsted et al., 2004). Edge effects typically arise when the wavelet is stretched and located near the start and end of the data because it extends outside the boundary of the time series. Therefore, information that lies outside the COI should be treated with caution.

Finally, we used WC to examine the strength and phase of the localised correlation between currents and wind stress. The arrows in WC analysis represent the relative phase or the time lag between both variables, with the level of statistical significance estimated using Monte Carlo methods (Grinsted et al., 2004). The time lag at a specific period is calculated as the phase angle divided by 2π times the period, though an anti-phase relationship could indicate negative correlation between the variables instead of one leading the other by half the period. We complemented WC with STHA to observe for any similar trends. We ran STHA through the MATLAB toolbox UTide (Codiga, 2011) using a 60-day window and moved with a time step of 1 day to provide seasonal resolution while maintaining result accuracy. A bimonthly window also resolves 35 tidal constituents, including those with longer periods up to monthly such as $M_m$, and in this case constituents with similar frequencies $N_2$ and $M_2$.

## 3 Results

### 3.1 General characteristics of currents and wind

Mean and maximum current speeds across the time-series was calculated for each site. Within the time domain, the mean speed of currents at the Pulau Hantu sites ranged from 9.3–19.6 cm s$^{-1}$ with maximum speeds of up to 1.2 m s$^{-1}$ (Table 1, Fig. 2). Currents at the Kusu Island sites are generally weaker throughout the duration with mean speeds ranging from 6.7–14.7 cm s$^{-1}$ (Table 1). Sites situated at the east and west of both islands experience higher mean speeds of 11.6–19.6 cm s$^{-1}$, as compared

to the north and south sites where mean speeds are weaker and do not exceed 10 cm s$^{-1}$ (Fig. 2). During the spring tides, current speeds more than quadrupled from about 20 cm s$^{-1}$ to about 1 m s$^{-1}$ consistently at both Hantu East and Hantu West. However, at Hantu North, Hantu South, and Kusu West, the increase in speed is less consistent, also reaching up to about 1 m s$^{-1}$, but only during 5 periods of spring tide every month during the monsoon seasons from May to September and from December to March. At Kusu South, currents exceeded 40 cm s$^{-1}$ only in Mar–Apr 2018 and remain weak over time. Kusu East and Kusu North exhibit consistent temporal variation, with the former recording slightly greater speeds during the NE monsoon.

**Table 1.** Properties of the principal components of both currents and wind stress (in italics).

| **Site** | **PC1 (%)** | **PC1 positive direction** | **PC2 positive direction** | **Mean speed (cm s$^{-1}$)** | **Direction of mean current speed (°)** | **Max current speed (cm s$^{-1}$)** |
|---|---|---|---|---|---|---|
| **Pulau Hantu** | | | | | | |
| North | 82.7 | SE | NE | 9.3 | 143.2 | 109.0 |
| South | 68.4 | NE | NW | 9.7 | 288.1 | 115.4 |
| East | 67.2 | NE | NW | 19.6 | 253.3 | 119.8 |
| West | 53.8 | NW | NE | 12.5 | 38.9 | 119.8 |
| *Wind Stress* | *60.7* | *NE* | *NW* | *0.0149 Pa* | *280.4* | |
| **Kusu Island** | | | | | | |
| North | 87.1 | SE | NE | 7.9 | 274.4 | 34.1 |
| South | 85.0 | NW | NE | 6.7 | 143.5 | 90.1 |
| East | 85.9 | NW | NE | 14.7 | 160.9 | 62.8 |
| West | 74.7 | NW | NE | 11.6 | 325.7 | 119.6 |
| *Wind Stress* | *76.9* | *NE* | *NW* | *0.0296 Pa* | *252.5* | |

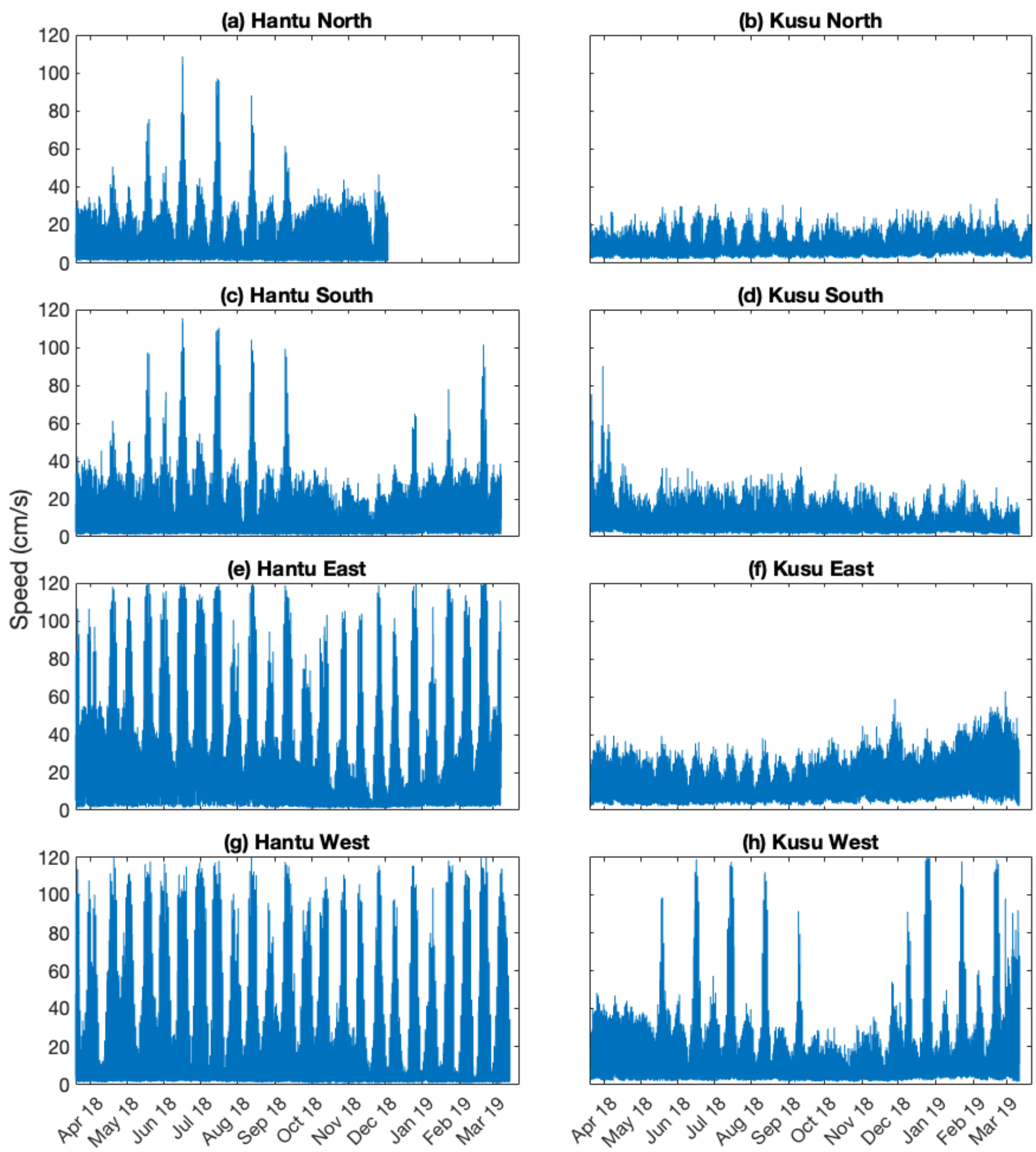

**Figure 2.** Current speeds over time at (a) Hantu North, (b) Kusu North, (c) Hantu South, (d) Kusu South, (e) Hantu East, (f) Kusu East, (g) Hantu West and (h) Kusu West.

PCA decomposition was applied to obtain the alongshore and cross-shore currents for each site. The alongshore current accounts for 67 % to 87 % of current variance in most study sites but only 54 % in Hantu West (Table 1). This could be attributed to the slightly concave nature of the coastline of Hantu West and hence flow is not strictly rectilinear. Meanwhile, the major axis winds over Pulau Hantu and Kusu Island account for 61 % and 77 % of the wind variance respectively and blow predominantly to the southeast from the South China Sea, with Kusu Island recording stronger winds. These winds strengthen during December to March and are thus representative of the NE monsoon. In contrast, the wind direction of minor axis winds is directed northwestward and become stronger from June to September, thereby coinciding with the SW monsoon. The wind rose for Pulau Hantu demonstrates the monsoon seasons experienced in Singapore (Fig. 3a).

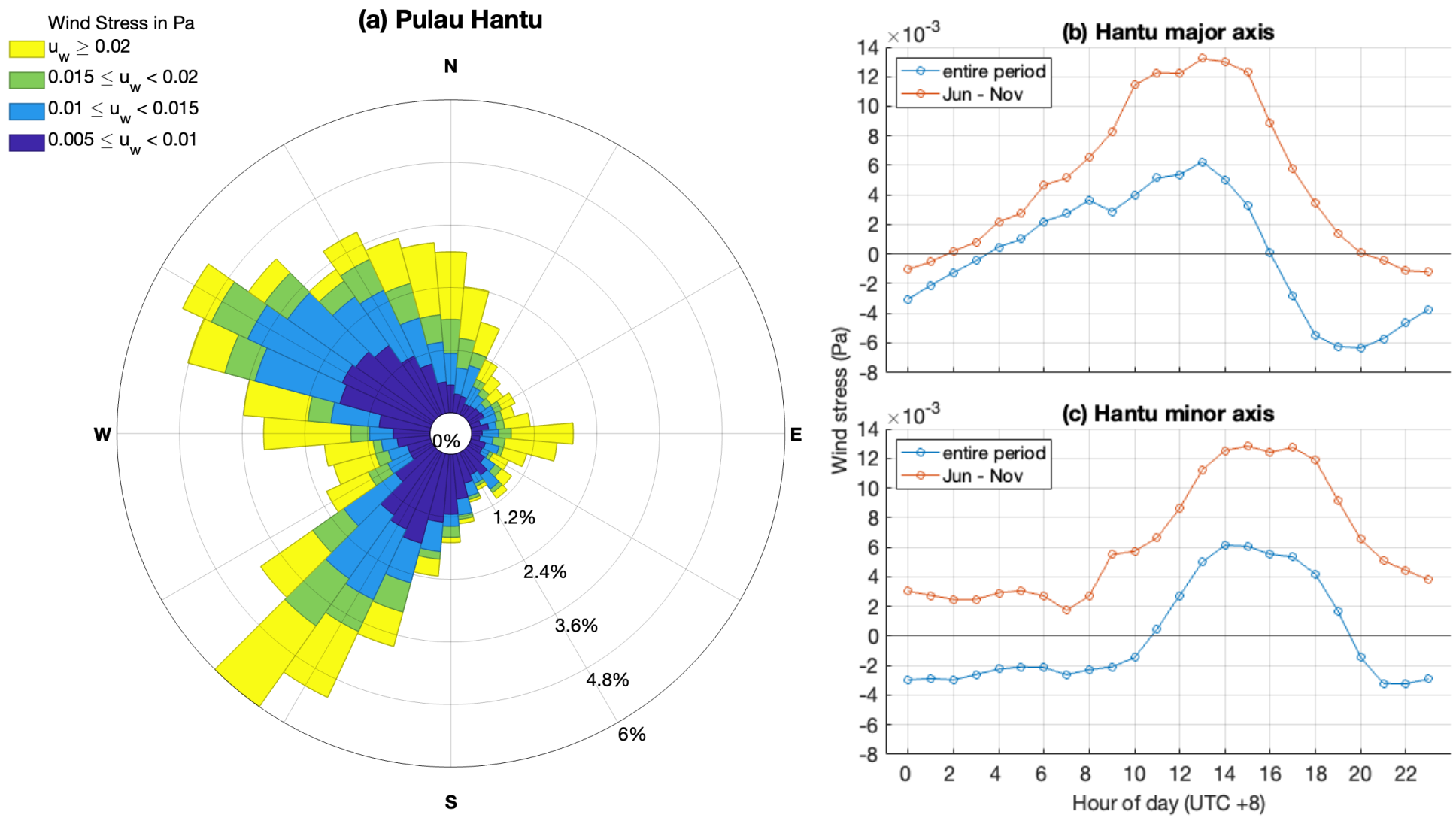

**Figure 3.** (a) Wind rose of Pulau Hantu from March 2018 to March 2019; and wind stress of Pulau Hantu averaged over hour of day, over the entire duration (blue) and from June to November (orange) for the (b) major axis, and (c) minor axis.

Wind stress was averaged for both Pulau Hantu and Kusu Island over each hour of the day (Fig. 3b). The average wind stress along both principal components is diurnal and asymmetric, characteristic of the land-sea breeze circulation in Singapore. The northeast and northwest directions are taken to be positive for the major and minor axis respectively (Table 1). During the day, averaged wind stress is directed north and reverses in direction at night, with maximum magnitudes of the major axis winds occurring at about 13:00 and 20:00 local time respectively. Owing to differential heating between the land and sea, the sea breeze blowing northward from the Singapore Strait during daytime is usually stronger than the land breeze coming from the Malay Peninsula late at night (Li et al., 2016). Interestingly, the diurnal pattern is more pronounced from June to November, which comprises of the SW monsoon and the following inter-monsoon period, where the averaged wind stress is directed northward at almost all hours of the day and increases in strength during the day. Likewise, we can infer the daily variation of averaged wind stress during the NE monsoon is a predominantly southward orientation that is most intense at night.

### 3.2 Power spectral density

We calculated the PSD to partition the variance into frequency bands and quantified them for both alongshore and cross-shore currents. Here we can observe that spectral peaks are most prominent at tidal frequencies, with diurnal currents being slightly stronger than semidiurnal currents (Fig. 4). The spectra yield two distinct peaks within the diurnal band, which correspond to the major diurnal tidal constituents $K_1$ (1.0032 cpd) and $O_1$ (0.9288 cpd). Higher frequency signals from the third diurnal to the eighth diurnal bands are also discernible, though not as strong as currents oscillating at tidal frequencies. Meanwhile, only the fortnightly signal is significant within the low-frequency band, which could represent the lunisolar synodic fortnightly tidal

constituent $MS_f$. The $MS_f$ constituent illustrates spring–neap variations caused by quadratic nonlinear interactions of the $M_2$ and $S_2$ tidal constituents (Hoitink and Jay, 2016). Wind stress spectra also show the presence of a strong land–sea breeze, evidenced by the sharp and narrow peaks occurring at periods of about 24 and 12 hours, the latter to a smaller extent.

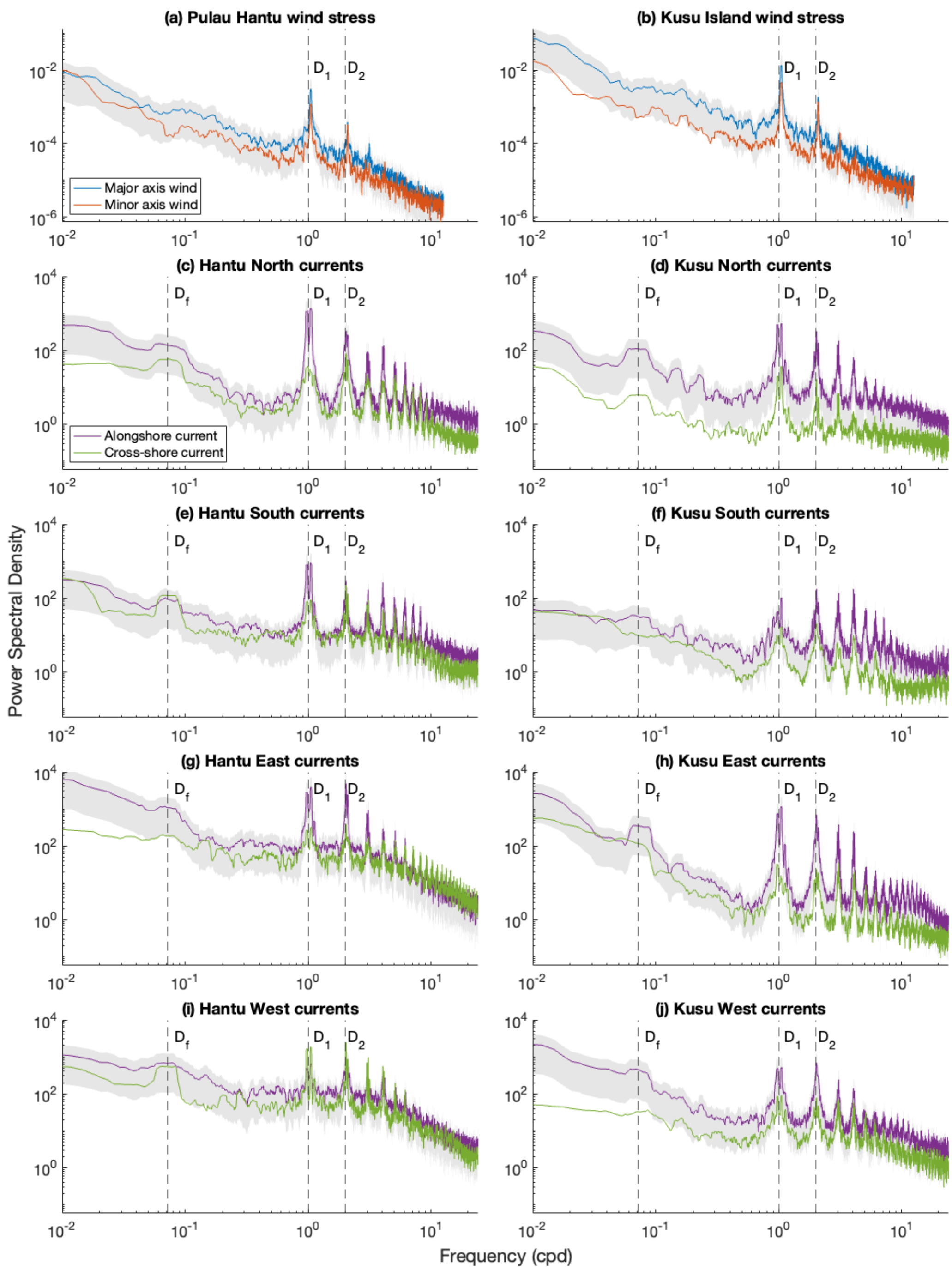

**Figure 4.** Power spectral density estimates of both major and minor axes wind stress in (a) Pulau Hantu and (b) Kusu Island, with units of $Pa^2$ $cpd^{-1}$, as well as that of both alongshore and cross-shore currents recorded at (c) Hantu North, (d) Kusu North, (e) Hantu South, (f) Kusu South, (g) Hantu East, (h) Kusu East, (i) Hantu West and (j) Kusu West, with units of $cm^2$ $s^{-2}$ $cpd^{-1}$. The shading indicates the 95 % confidence level for the major axis wind stress and alongshore currents. The vertical dashed black lines represent (from left) the fortnightly ($D_f$) shown only in the currents plots, diurnal ($D_1$), and semidiurnal ($D_2$) frequency bands.

Table 2 presents the quantification of variance for each frequency band. The computed variance of alongshore and cross-shore currents, denoted by PC1 and PC2 respectively, are similar to the PCA results. Tidal frequencies are responsible for about 26–45 % and 14–36 % of the total current variance in Pulau Hantu and Kusu Island respectively, with diurnal currents being stronger than semidiurnal currents by a factor of at least 1.3 in Hantu North, Hantu North, Kusu East, and Kusu South (Table 2). High frequency motions, which consist of mainly overtides, are depicted in the peaks from the third to eighth diurnal frequency bands, which further attest to the chaotic behaviour of tidal motions occurring near the coasts.

**Table 2.** Variance (in $cm^2$ $s^{-2}$) of the filtered currents obtained from PSD.

| **Site** | **Frequency band** | | | | | | | | | | | |
|---|---|---|---|---|---|---|---|---|---|---|---|---|
| | **Low-frequency** | | | **Diurnal** | | | **Semidiurnal** | | | **Total** | | |
| | **PC1** | **PC2** | **Total** | **PC1** | **PC2** | **Total** | **PC1** | **PC2** | **Total** | **PC1** | **PC2** | **Total** |
| **Pulau Hantu** | | | | | | | | | | | | |
| North | 26.9 | 6.7 | 33.6 | 126.2 | 7.6 | 133.8 | 49.0 | 10.6 | 59.6 | 357.4 | 73.7 | 431.1 |
| % | 6.2 | 1.6 | 7.8 | 29.3 | 1.8 | 31.0 | 11.4 | 2.5 | 13.8 | 82.9 | 17.1 | 100.0 |
| South | 23.0 | 18.1 | 41.1 | 77.9 | 15.5 | 93.4 | 30.8 | 19.3 | 50.1 | 400.3 | 162.4 | 562.7 |
| % | 4.1 | 3.2 | 7.3 | 13.8 | 2.8 | 16.6 | 5.5 | 3.4 | 8.9 | 71.1 | 28.9 | 100.0 |
| East | 289.8 | 53.0 | 342.8 | 363.9 | 67.1 | 431.0 | 402.1 | 68.6 | 470.7 | 1567.6 | 616.6 | 2184.2 |
| % | 13.3 | 2.4 | 15.7 | 16.7 | 3.1 | 19.7 | 18.4 | 3.1 | 21.6 | 71.8 | 28.2 | 100.0 |
| West | 150.5 | 67.2 | 217.7 | 131.1 | 159.7 | 290.8 | 178.9 | 160.1 | 339.0 | 1081.5 | 809.0 | 1890.5 |
| % | 8.0 | 3.6 | 11.5 | 6.9 | 8.4 | 15.4 | 9.5 | 8.5 | 17.9 | 57.2 | 42.8 | 100.0 |
| **Kusu Island** | | | | | | | | | | | | |
| North | 17.9 | 1.7 | 19.6 | 48.9 | 2.9 | 51.8 | 29.7 | 1.5 | 31.2 | 201.1 | 28.9 | 230.0 |
| % | 7.8 | 0.7 | 8.5 | 21.3 | 1.3 | 22.5 | 12.9 | 0.7 | 13.6 | 87.4 | 12.6 | 100.0 |
| South | 6.8 | 3.7 | 10.5 | 11.3 | 2.6 | 13.9 | 17.1 | 3.8 | 20.9 | 202.0 | 45.5 | 247.5 |
| % | 2.7 | 1.5 | 4.2 | 4.6 | 1.1 | 5.6 | 6.9 | 1.5 | 8.4 | 81.6 | 18.4 | 100.0 |

| | | | | | | | | | | | | |
|---|---|---|---|---|---|---|---|---|---|---|---|---|
| East | 76.5 | 24.0 | 100.5 | 87.6 | 4.2 | 91.8 | 64.1 | 3.5 | 67.6 | 374.6 | 64.8 | 439.4 |
| % | 17.4 | 5.5 | 22.9 | 19.9 | 1.0 | 20.9 | 14.6 | 0.8 | 15.4 | 85.3 | 14.7 | 100.0 |
| West | 91.6 | 8.4 | 100.0 | 74.7 | 14.8 | 89.5 | 92.2 | 13.2 | 105.4 | 601.3 | 165.2 | 766.5 |
| % | 12.0 | 1.1 | 13.0 | 9.7 | 1.9 | 11.7 | 12.0 | 1.7 | 13.8 | 78.4 | 21.6 | 100.0 |

### 3.3 Continuous wavelet transform of currents and wind stress spectral density

The CWT of alongshore and cross-shore currents at all study sites show that the diurnal and fortnightly signals are strongest and statistically significant (Fig. 5 and Fig. S1 in Supplementary material). In addition, energetic semidiurnal and monthly signals are also present. Strong alongshore currents at tidal frequencies occur about twice a month, which incidentally coincided with the spring–neap tidal cycle. The presence of statistically significant regions in Hantu West is better observed in the CWT of its cross-shore currents (Fig. S1), where there is minimal spectral leakage and currents are strongest at tidal and fortnightly frequencies. High frequency oscillations are also present all-year round, highlighting the chaotic nature of currents flowing near the coasts. The wavelet power is generally negligible between the period of 1 to 8 days.

Nonstationary behaviour of currents is clearly exhibited as the wavelet power across frequency bands show significant temporal variation. Alongshore currents are unusually weak during the inter-monsoon period from September to October in Hantu East, Hantu South and Kusu West, with Kusu West showing the greatest contrast (Fig. 5). At Kusu East and Kusu North, diurnal currents seem to intensify when the NE monsoon begins in December, which contrasts with Kusu South where currents at tidal frequencies are heavily attenuated from November onwards (Fig. 5).

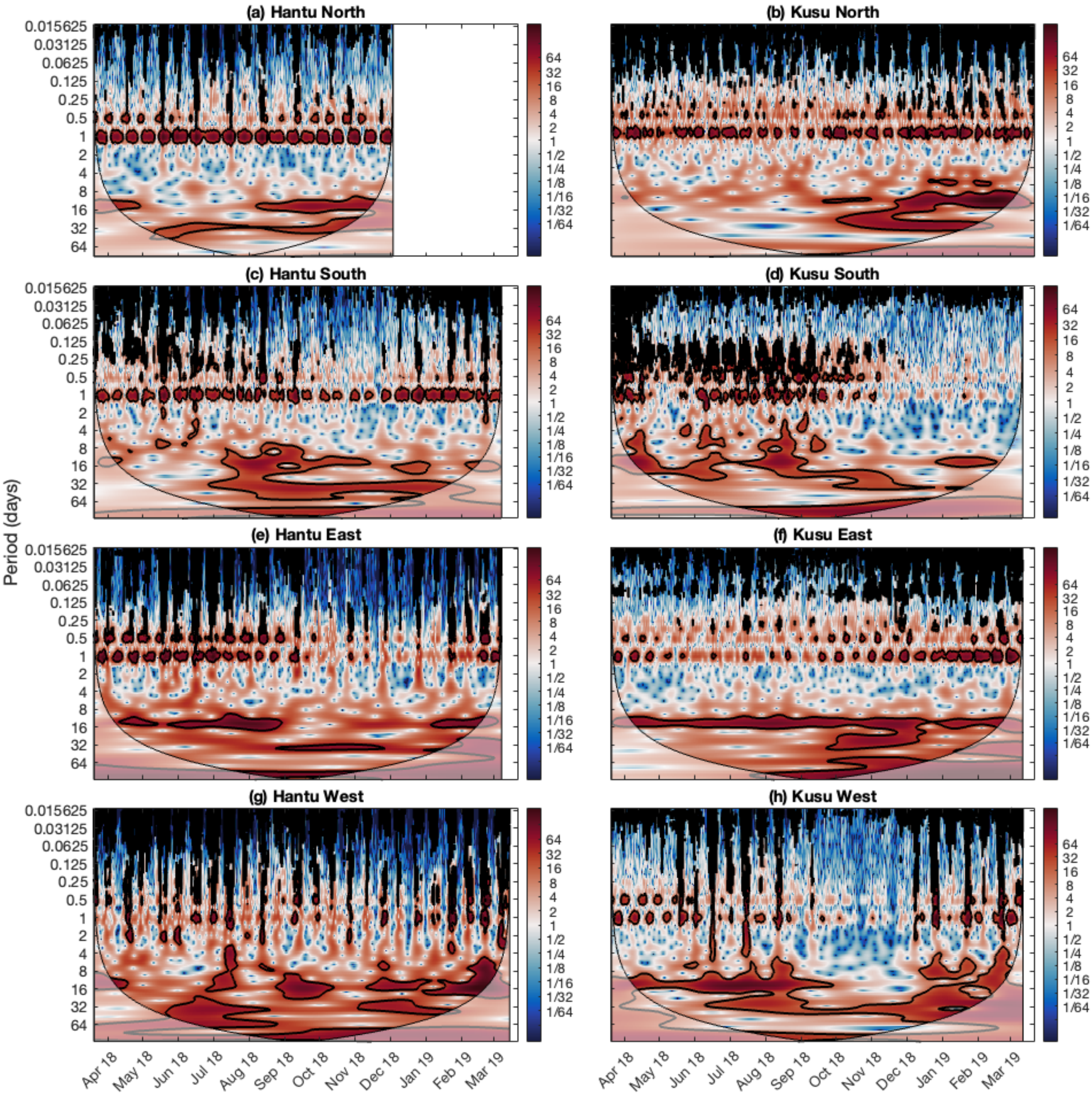

**Figure 5.** CWT of alongshore currents at (a) Hantu North, (b) Kusu North, (c) Hantu South, (d) Kusu South, (e) Hantu East, (f) Kusu East, (g) Hantu West and (h) Kusu West. The thick black contour encompasses significant regions against red noise ($p < 0.05$), and the cone of influence (COI) is shown as a lighter shade where edge effects cannot be ignored. The wavelet power is $\log_2(A^2\ v^{-1})$, where A is the wavelet amplitude, and v is the variance of the entire signal.

The CWT of wind stress present powerful and statistically significant oscillations at the diurnal and monthly frequencies (Fig. 6). Diurnal wind stress is strongest during the NE monsoon from December to March. The minor axis diurnal wind stress at Pulau Hantu, while not as strong during the NE monsoon, recorded considerable strength in the months from June to October, the period when the SW monsoon regime is dominant. At subtidal frequencies, power is most concentrated for periods of about 32 days from December to February for Pulau Hantu, and from April to October for Kusu Island. Similar to the CWT for currents, significant regions located outside the COI are not as reliable and thus ignored.

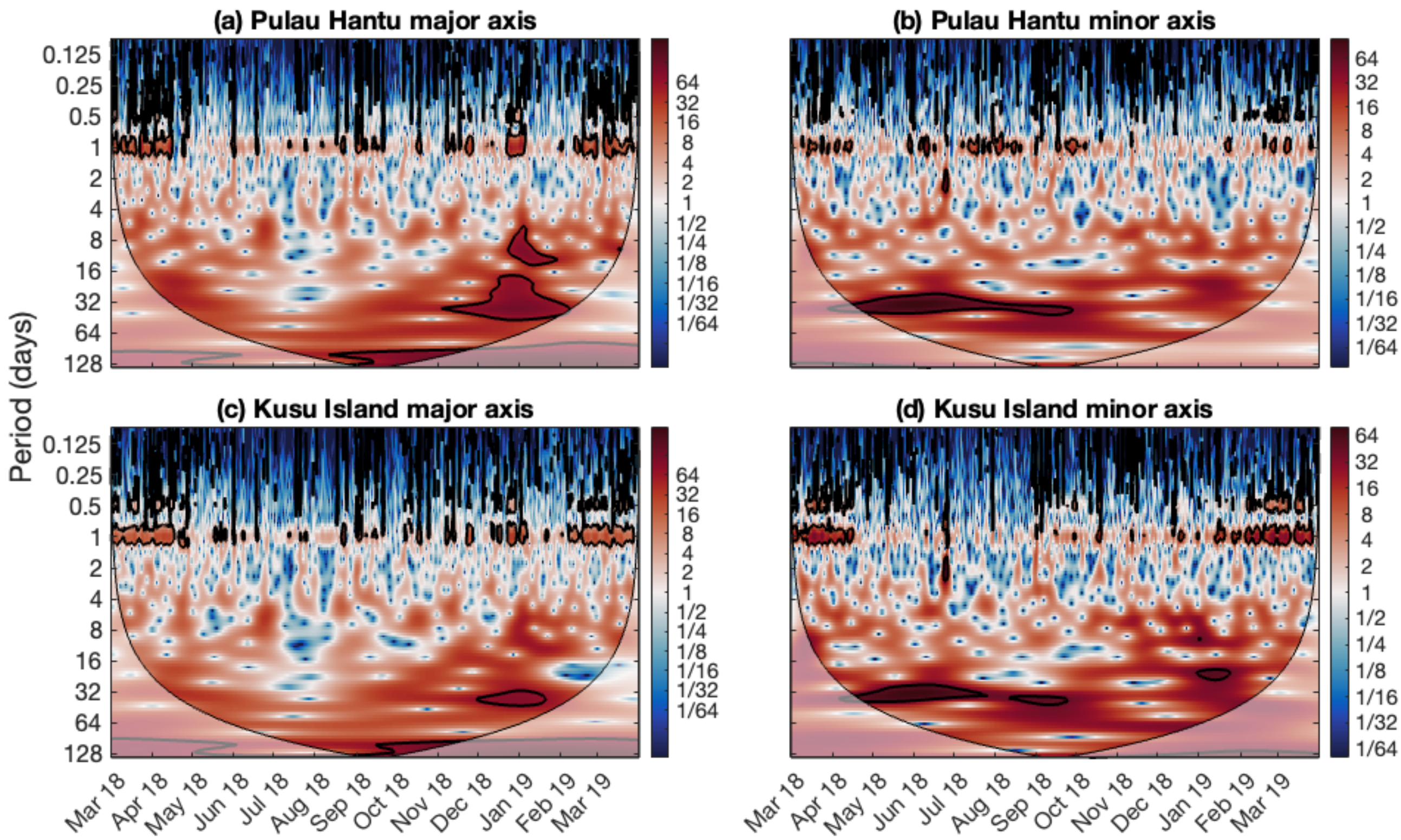

**Figure 6.** CWT of wind stress at (a) Pulau Hantu major axis, (b) Pulau Hantu minor axis, (c) Kusu Island major axis, (d) Kusu Island minor axis.

### 3.4 Wavelet coherence between currents and wind stress, and STHA of currents

The WC between both alongshore currents (Fig. 7) and cross-shore currents (Fig. S2 in Supplementary material) and major axis wind stress reveal strong correlation over a broad range of frequencies, with the subtidal and diurnal frequency bands being the most significant. Correlation from September onwards is high, significant, and largely in phase at the monthly frequency, though some areas are suspect as they lie outside the COI. At Hantu North and Kusu East, the relative phase is approximately 90–120° pointing down at the monthly frequency, indicating that strong currents occur about 1 to 2 weeks later.

At the diurnal frequency band, correlation was generally significant, albeit rather sporadically, during the monsoon seasons for most sites. This suggests that nontidal processes such as wind stress introduces nontidal energy at tidal frequencies (Pugh, 1987). In addition, the relative phase relationship has considerable temporal and spatial variability. Correlation is weak and close to zero during the inter-monsoon from September to November, albeit with short periods of high correlation in between (Fig. 7 and Fig. S2). During the SW monsoon from June to September, correlation between the major axis wind stress and alongshore currents is high and approximately in phase at Hantu East, Hantu South, Kusu East and Kusu North (Fig. 7), with a phase angle at Kusu East of about 0–90°. This corresponds to wind stress leading currents with a time lag of up to 6 hours. In contrast, the correlation at Hantu North is almost anti-phase. Considering the NE monsoon, statistically significant regions are observed in Hantu East, South, Kusu East, North and West. The correlation is approximately in-phase at Kusu West and anti-phase at Hantu East and Kusu East, while the phase relationship at Hantu South and Kusu North is highly variable with the phase angle fluctuating wildly from 0–270°. This converts to a time lag of up to 6 hours with respect to both the phase and antiphase.

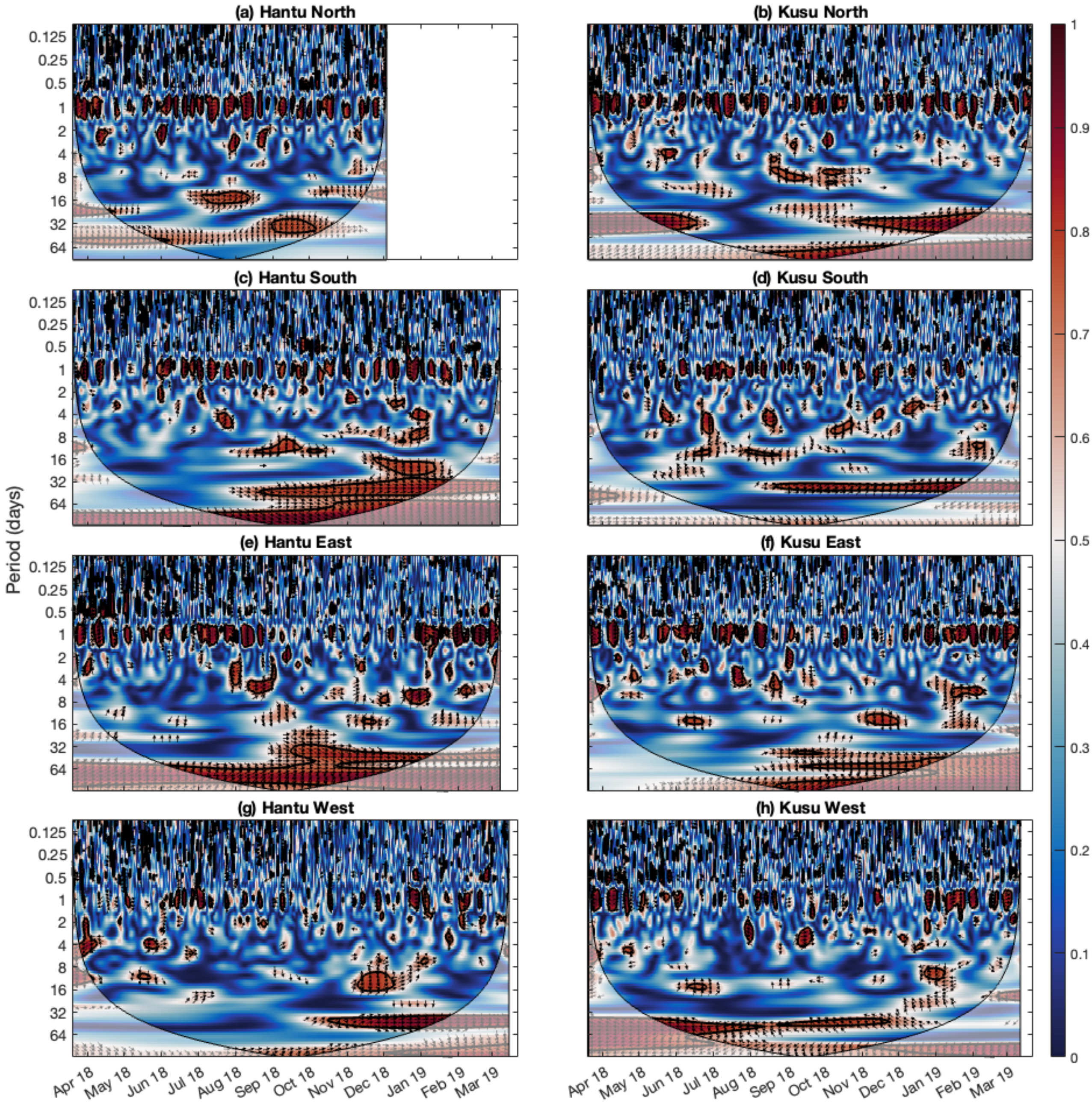

290 **Figure 7.** Wavelet coherence between alongshore currents and major axis wind stress at (a) Hantu North, (b) Kusu North, (c) Hantu South, (d) Kusu South, (e) Hantu East, (f) Kusu East, (g) Hantu West and (h) Kusu West. The black contours indicate significant regions and the cone of influence (COI). The area that lies outside the COI has a lighter shade and information in this area should be treated with caution. The arrows represent the relative phase relationship, with arrows pointing right and left indicating in-phase and antiphase respectively, and winds leading currents by 90° pointing down.

We presented the semi-major axis amplitude of four main tidal constituents $K_1$ (Fig. 8), $O_1$, $M_2$, and $S_2$ (Figs. S3–S5 respectively in Supplementary material). The amplitudes of most major tidal constituents do not seem to exhibit any trend. However, except for Kusu South where currents are generally very weak, the amplitude of the $K_1$ tidal constituent appeared to vary semi-annually and ranges from about 1 to 18 cm $s^{-1}$ during the monsoon seasons. In Pulau Hantu, the $K_1$ amplitude is generally greater than that in Kusu Island, which is not surprising since currents in Pulau Hantu are generally stronger. This suggests that diurnal wind forcing amplified during the monsoon seasons results in a corresponding increase of the $K_1$ amplitude of currents.

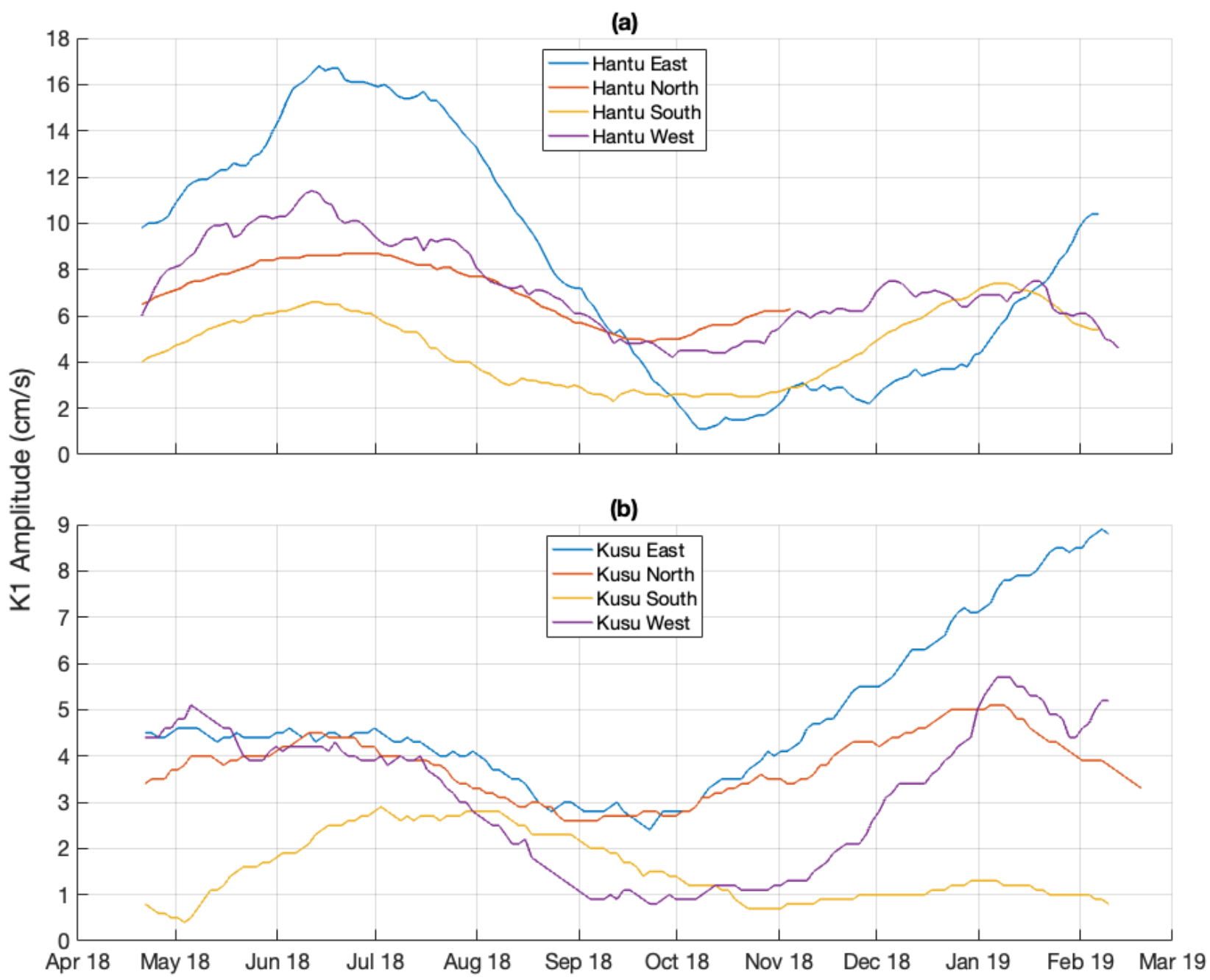

**Figure 8.** $K_1$ tidal amplitudes of all sites in (a) Pulau Hantu and (b) Kusu Island, derived from STHA.

## 4 Discussion

The high correlation displayed in the coherence analysis indicates the significance of wind forcing in driving currents. During the SW monsoon, the effect of wind forcing at Hantu East, Hantu South, and Kusu North is immediate as implied by the in-phase relationship. We speculate that as both the coastline and the major axis wind stress are oriented northeastward, they are more aligned with each other, thereby facilitating current flow along that direction. However, Hantu North and Kusu East sites are oriented northwestward and hence the response to wind forcing is not immediate. Additionally, the anti-phase relationship observed in Hantu North implies negative correlation, which could be attributed to the currents flowing southward facing resistance from the local winds blowing north and thus weakening in strength.

The results of STHA could offer clues to the seasonal patterns observed in the WC analysis. Though we expect coastal currents to exhibit strong seasonal variation in tandem with the monsoon seasons, the CWT results showed that diurnal currents remain dominant for the entire observational period. Hence, we speculate that the change in velocity of diurnal currents due to wind stress is only marginal. Pugh (1987) documented an amphidromic point for the main diurnal constituents in the Singapore Strait that is responsible for strong diurnal currents. Van Maren and Gerritsen (2012) further deduced in their study that the $K_1$ tidal wave is a standing wave as it propagates from the South China Sea and reflects off the Sumatra coast, generating high velocities in the Singapore Strait that are predominantly diurnal despite Singapore experiencing a semidiurnal tidal regime. Moreover, the semi-annual variation of the $K_1$ amplitude demonstrates the pivotal role of the monsoons in intensifying the land–sea breeze and subsequently enhance the astronomical $K_1$ tidal forcing. This result corroborates with prior results presented by Álvarez et al. (2003) and Dusek et al. (2017), both of whom concluded that the strength of sea breeze wind stress is responsible for the seasonal variations of the $K_1$ amplitude of estuarine tidal currents in Cádiz Bay, Spain, and Tampa Bay, Florida, respectively.

The amplification of diurnal currents in response to the land–sea breeze has wider implications for nearshore sediment dynamics and coral reef ecosystems in the Singapore Strait. Stronger currents create higher shear stress at the seabed and are thus responsible for the increased resuspension of fine sediment (Fearon et al., 2020). In western islands such as Pulau Hantu where diurnal currents are stronger, coral reefs become more vulnerable to the effects of higher turbidity due to the decrease in available habitat for growth (Morgan et al., 2020). Meanwhile, Kusu Island is reported to be one of the most robust sources of coral larvae seeding as it is situated upstream from the dominant current flowing net westward in the Singapore Strait (Tay et al., 2012). Given how coral reef larvae are heavily reliant on currents for dispersal (Tay et al., 2012), a deepened understanding of surface current variability will help reef management adopt targeted measures in conserving coastal areas with high seeding potential. Careful attention should also be given to the response of coral reef larvae to increased wind speeds, particularly since the localised high correlation between currents and winds during the monsoon seasons can be observed from the WC analysis. The wavelet techniques employed here can not only provide insights to time dependent dynamics of coastal currents, but also in other oceanographic processes not limited to coral larvae transport as mentioned earlier, sediment dynamics, and internal tides.

Lastly, the findings of spectral analysis underscore the importance of obtaining longer time-series, especially since regions at subtidal frequencies in the CWT and WC plots are strong and significant. However, we are constrained by the length of observational data and are thus only able to analyse signals with periods of up to 2 months, any period longer than that rendered unreliable due to edge effects that arise when performing wavelet analysis of low frequency signals. With longer datasets, we can better determine the effects of not only monsoons but other atmospheric forcing occurring at longer time scales such as the El Nino Southern Oscillation on currents. Such events are projected to intensify due to the increase in land–sea temperature

contrast associated with climate change (Kitoh, 2017), with considerable changes in the strength of currents (Sen Gupta et al., 2021).

## 5 Conclusions

In this study, we investigated the nonstationary behaviour of coastal currents and their main drivers in Singapore. Despite the study sites being in close proximity, the variations in speed, PSD, CWT of alongshore currents, and WC between that and major axis winds revealed substantial heterogeneity across time and space. Current speeds in Hantu East and Hantu West consistently exceeded 1 m $s^{-1}$ during spring tides, while that in Hantu North, Hantu South and Kusu West only exceeded 1 m $s^{-1}$ in spring tides monthly during the monsoon seasons. Currents in the rest of the sites in Kusu Island are generally weak. Tides and winds are the two dominant mechanisms, the former accounting for 14–45 % of the total variance of currents for all study sites from the PSD plots. CWT not only displayed high power observed at tidal and fortnightly frequencies, but also elucidated the nonstationary behaviour as the strength of currents flowing at these frequencies exhibited temporal variation. Meanwhile, strong winds strengthen the diurnal land–sea breeze which in turn influence diurnal currents during the monsoon seasons. The high localised correlations can be observed in the WC plots of Hantu East, Hantu North, Hantu South, and Kusu East at the diurnal frequency. This could be explained by the seasonal variation of the $K_1$ amplitude of currents, where it increases during the onset of monsoonal winds and decreases during the inter-monsoons. Such results are crucial and should be considered in future hydrodynamic models in order to better predict coastal currents across different time scales.

## Code and data availability

ERA5 hourly wind data at 0.25 ° x 0.25 ° resolution, not downscaled, are publicly available and can be downloaded from https://cds.climate.copernicus.eu/cdsapp#!/dataset/reanalysis-era5-single-levels?tab=overview. The database is available through an unrestricted data repository (DR-NTU) hosted by Nanyang Technological University (https://doi.org/10.21979/N9/ICJXJ3) (Puah and Morgan, 2024).

## Author contributions

JYP performed all statistical analysis, developed the code, created the figures, and drafted the manuscript. IDS provided advice on paper structure. KM collected the observational data. DL, DP and MW provided general direction of the paper. ADS provided direction for conceptualisation. All authors contributed to the scientific discussion of methods and results, as well as the editing of the manuscript.

## Competing interests

The authors declare that they have no conflict of interest.

**Acknowledgements**

This Research/Project is supported by the National Research Foundation, Singapore, National Environment Agency, Singapore, under the National Sea Level Programme Funding Initiative (Award No. USS-IF-2020-2). This work comprises EOS contribution number 569. We also like to thank Dr Srivatsan Vijayaraghavan from Tropical Marine Science Institute for providing the downscaled wind data and the methods used in downscaling. Colormaps are from the cmocean package (Thyng et al., 2016).

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
