# Peer review of "Importance of tides and winds in influencing the nonstationary behaviour of coastal currents in offshore Singapore"

_EGUsphere, 2024_

## Author Comment (AC1)

**Answer to Anonymous Referee #1**

This manuscript analyzed one-year observed tidal currents around two islands in offshore Singapore, using the short-term harmonic analysis, continuous wavelet transform and, magnitude-squared wavelet coherence. The results reveal the importance of tidal motions and monsoons in diurnal and subtidal periods. This manuscript is reasonably organized and the results can support the main conclusions. Please consider the below comments for improving the manuscript.

We thank the referee for their kind words.

- The authors use the WRF model to downscale the winds to 10 km resolution, so wind data should be available for both Pulau Hantu and Kusu Island for comparison, may the authors update Figure 3 and have a look at if there is any difference in wind strength and directions, which are important for further exploring the correlation between wind and currents in next sections.
- We have plotted the wind rose for Kusu Island as well. We updated Figure 3 and overlay both wind roses onto the maps with the tidal ellipses.

- May the authors provide the co-tidal and co-range charts in the vicinity of Singapore? Maybe it can help analyze the variation of the tidal amplitude and corresponding reasons.
- Thank you for the suggestion. While we agree that co-tidal and co-range charts could help in analyzing the variation of the tidal amplitude and corresponding reasons, we prefer to focus on interpreting the results of wavelet analysis to explain the nonstationary behaviour of coastal currents.

---

## Author Comment (AC2)

**Answer to Anonymous Referee #3**

This manuscript presents a study of the spatiotemporal variability of coastal currents in shallow coral reef environments of southern Singapore. The authors employ a robust methodology, utilizing eight tilt current meters deployed over a one-year period to analyze current patterns in the frequency domain. The research makes several significant contributions to the field. It quantifies the relative importance of tidal motions in driving current flow, demonstrating that they account for 14-45% of total variance across sites. The finding that diurnal currents exhibit similar or greater energy proportions compared to semidiurnal currents is noteworthy and contributes to our understanding of local hydrodynamics. The authors apply wavelet coherence analysis to examine the relationship between wind stress and current. This approach reveals important correlations at diurnal and subtidal frequencies during monsoon periods, with observed time lags of up to 6 hours in both phase and antiphase.

Overall, this manuscript presents valuable findings that advance our understanding of coastal current dynamics in coral reef environments that have valuable implications for port activities, coastal landform morphodynamics, and ecosystem functioning. However, the manuscript can be improved further with another revision to better contextualize this study.

We thank the referee for their kind words.

Specific comments:

1. Include a regional map highlighting the area's importance (transit between Pacific and Indian Oceans; trade routes) and presenting regional winds and hydrodynamics.

   Figure 1 is updated to include a regional map of Southeast Asia with important labels that indicate Singapore being situated in between the South China Sea and the Strait of Malacca. The winds and currents are depicted as wind roses and tidal ellipses respectively in Figure 3 of the revised manuscript.

2. Lines 132-135: Provide more detail on the WRF model setup.

   We provided additional details on the WRF model in the last paragraph of section 2.1. We specified the model version, the grid domain which covers Southeast Asia, the horizontal grid resolution of 10 km, and the centre coordinates of the grid.

3. Consider the location of each data collection point in the results analysis. For example, Kusu South, North, and East sites have different wind patterns throughout the year, which may explain the contrast in current patterns between Kusu South and North (line 254). Averaging wind stress over the four areas presented in Figure 6 may filter out their variabilities.

The downscaled wind data has a spatial resolution of 10 km. The scale bar in Figure 1 provides a good visualisation of the spatial scale, and from Figure 1 it can be estimated that both Pulau Hantu and Kusu Island measured from east to west is no longer than 1 km. As such, the wind rose for Kusu Island can reasonably be assumed to represent the wind patterns for all sites within Kusu Island, and similarly for Pulau Hantu.

Nevertheless, we acknowledge that current patterns within Kusu Island are different and we consider the local bathymetry as a possible factor behind these differences.

4. Lines 234-237: Clarify how the percentages 26-45% and 14-36% were derived in Table 2.

We mentioned in section 2.2 that the quantification of variance was done by integrating the spectrum over each frequency band and assessed their relative contribution. We have now added the method of integration, which is rectangle approximation, and the expression of the relative contribution as a percentage of the total variance, to the main text for clarity. Additionally, we have also added the explanation to the caption of Table 3 in the revised manuscript.

Technical comments:

1. Line 40: Clarify whether current ellipse parameters are estimated in this study or can be estimated in general.

Current ellipse parameters can be estimated in general using harmonic analysis, and the estimation is done in this study as described in section 2.2 of the revised manuscript primarily based upon feedback from Reviewer 2. The ellipses for major tidal constituents are then plotted in Figure 3 of the revised manuscript.

2. Line 236: Remove the duplicate mention of Hantu North.

Thank you for spotting this duplicate error. The correct sites should be Hantu North, Hantu South, Kusu North and Kusu East.

3. Figures 5 and 7: Confirm whether the Y-axis represents period or frequency.

The Y-axis represents period, already labelled as "Period (days)" on the left side of both figures.

---

## Author Comment (AC3)

**Answer to Anonymous Referee #2**

This manuscript presents a detailed study of the influence of tides and wind on tidal currents offshore Singapore, which was developed using a series of statistical methods. Careful analysis with solid data recorded (1 year) from fieldwork is presented, and the authors expose this detailed combination of statistical processes to assess the non-stationary behavior of coastal currents.

While I found the authors have done excellent work and the paper is well-written, I still suggest a moderate revision following the comments that I listed below:

We thank the referee for their kind words.

1. The authors need a map or more to show the directions of minor and major axes of the tidal current (e.g., tidal ellipse) at each TCM, the direction of wind to explain the conclusions. As a reader, I found it difficult to figure out the direction of wind stress, tidal currents, and the coastline.

   We have updated Figure 1 to include the bathymetry in the map, as well as Figure 3. Figure 3 now consists of zoomed-in maps of Pulau Hantu and Kusu Island with their wind roses and tidal ellipses of $M_2$, $S_2$, $K_1$, and $O_1$ at each TCM. The tidal ellipse parameters are calculated by performing harmonic analysis over the entire observation period.

2. The wavelet analyses (CWT and WC) are really insightful in terms of the results. The non-stationary behaviors of coastal currents can be seen clearly. However, in my point of view, the authors should discuss this outcome as a "low-frequency signal". That is more meaningful as the authors also mentioned ENSO in their discussion. For low-frequency signals, as the wavelet indicates some period of 32 days (L266), I suggest the authors explore the linkage to the intraseasonal variability (30 to 90-day) in the tropical atmosphere (e.g., Madden-Julian Oscillation)

   We thank the referee for making this additional point. For a start, we redefined such signals with longer periods as "low-frequency signals" instead of "signals with subtidal frequencies". We briefly introduce the MJO as a dominant influence of intraseasonal atmospheric variability and expanded on its characteristics and relationship with low-frequency coastal currents in the Discussion section.

3. Lines 275-280: The tidal currents are also distorted by the changes in bathymetry. Can the authors discuss the role of bathymetry more here or provide more information about bathymetry on this study site?

   We briefly described the bathymetry in the Introduction and delved deeper into how the complex bathymetry distorts the tidal currents, as seen from the fluctuating phase relationship with the diurnal wind stress over the entire observation period, in the Discussion section. The deeper depths observed near Kusu Island likely help to explain the unusually weak tidal currents in Kusu South.

4. I need clarification when the authors discuss the seasonal variation of the K1 amplitude of currents. The authors discuss that this might be due to the monsoonal winds. However, the results show that other major tidal constituents (O1, M2, and S2) are also large, but they did not present seasonal variation as K1 (Supp. Material). So why do coastal currents only follow K1? I think the authors need to discuss the interaction between tidal constituents in more detail here (e.g., in Van Maren and Gerritsen, 2012). The characteristics of the tide in the SCS are complicated as the oscillations of standing waves could produce resonance in a semi-closed or complex coastline water body. I suggest the authors put more analyses or explanations in place to make this point more solid.

Although performing harmonic analysis over the entire observation period ignores non-stationary behaviour, we are nevertheless able to extract additional tidal constituents that could be useful in explaining the seasonal variation. In the Discussion section, we explored the result of the interaction between $P_1$ and $K_1$, the former which cannot be resolved using STHA, and discovered that this interaction gives rise to the semi-annual variation of both the strength of alongshore currents and also possibly the $K_1$ amplitude. This interaction is depicted in Figure 10 of the revised manuscript.

Other minor comments:

- Fig.7: it is really hard to see the arrows on the graph. Also, a legend of a vertical color bar needed to be included.
- We have replaced the diverging colourmap, which in hindsight is better suited for displaying anomalies, with a sequential colourmap for all wavelet analyses. The sequential colourmap highlights regions with higher values yellow, which contrasts well with the black arrows in Figure 7, now relabelled as Figure 8 in the revised manuscript. We have also made the arrows slightly bigger and labelled the colour bar.

- L175: need to make the unit consistent throughout the manuscript
- Thank you for pointing this out. We have changed the unit to cm s$^{-1}$ throughout the manuscript.

- L284 and L288: The phase angles are 0-90 and 0-270, respectively, but why are they converted to a same time lag of up to 6 hours? Are they the same, or do we have a time lag and time lead here?
- We have added extra explanations on the interpretation of the phase angles in sections 2.2 and 3.5, and all phase angles were calculated with respect to the phase. We also specified the leading variable depending on the direction of the arrows.

- Table 1: How much is the percentage of explained variance for PC2 at each site? As they kept PC2 in, they need to provide the percentage as they did for PC1.

- We have added the column of the percentage of explained variance for PC2. For clarity, we have also split Table 1 into two tables (Tables 1 and 2 in the revised manuscript) showing properties of currents and wind separately.

- Table 1: Please check the header: "Direction of mean current speed"? Or just "Direction of mean current?"
- Thank you for spotting this. We have changed the header to "Direction of mean current".

- L193: As I commented above, the authors need a map to show the direction of the coastline and the currents.
- We have corrected this, please see reply to the above comment.

- Fig 3: I suggest the authors show a graph of monthly averaged wind stress for the study site.
- Good suggestion. We removed the hourly averaged wind from June–Nov and combine all the plots for Hantu and Kusu into one figure (Figure 4a in the revised manuscript). We did the same for monthly averaged wind as well (Figure 4b in the revised manuscript).

- L328-340: I think this paragraph is better to be in the conclusion rather than in the discussion
- Good point, thank you. We have moved the paragraph to the conclusion.

---

## Author Response (AR2)

Dear Editor,

Please find enclosed the point-by-point responses to the referees' comments on the manuscript.

Best regards,

Jun Yu Puah and co-authors

Can the authors provide information on bathymetry data used to plot Fig1?

In Section 2.1, we added a new paragraph in lines 122–127 of the revised manuscript describing the bathymetry data and the sources. In the Figure 1 caption, we also added in-text citations that were used to plot the regional map and the bathymetry background colour. We also renamed the heading of Section 2.1 from "Field data collection" to "Data Collection".

Can the authors explain how they calculated in Table 3? I do not understand how they came up with the numbers in the Total column. Is it not the sum of, for example, PC1 in Low - Frequency, Diurnal and Semi-diurnal?

We have made changes in Table 3 and the paragraph preceding Table 3 for clarity. The main highlight is that the total variance is the sum of the variances of low-frequency (LF), diurnal (D), semidiurnal (SD), and high-frequency (HF) bands. The abbreviations were also included in Table 3. We also emphasised in the paragraph and the table caption that Table 3 only included information on the LF, D, and SD frequency bands, and the total variance, while information on the HF frequency band is excluded from the table.